# Treatment Patterns and Outcomes by Age in Metastatic Urinary Tract Cancer: A Retrospective Tertiary Cancer Center Analysis

**DOI:** 10.3390/cancers16112143

**Published:** 2024-06-05

**Authors:** Nishita Tripathi, Georges Gebrael, Beverly Chigarira, Kamal Kant Sahu, Ishwarya Balasubramanian, Constance Caparas, Vinay Mathew Thomas, Jessica N. Cohan, Kaitlyn Pelletier, Benjamin L. Maughan, Neeraj Agarwal, Umang Swami, Sumati Gupta

**Affiliations:** 1Huntsman Cancer Institute, University of Utah, Salt Lake City, UT 84112, USA; nishita.tripathi@hci.utah.edu (N.T.); georges.gebrael@hci.utah.edu (G.G.); beverly.chigarira@hci.utah.edu (B.C.); kamal.sahu@hci.utah.edu (K.K.S.); ishwarya.bala@utah.edu (I.B.); constance.caparas@hci.utah.edu (C.C.); vinay.mathewthomas@hci.utah.edu (V.M.T.); jessica.cohan@hsc.utah.edu (J.N.C.); kaitlyn.pelletier@hci.utah.edu (K.P.); benjamin.maughan@hci.utah.edu (B.L.M.); neeraj.agarwal@hci.utah.edu (N.A.); umang.swami@hci.utah.edu (U.S.); 2Detroit Medical Center, Wayne State University, Detroit, MI 48201, USA; 3George E. Wahlen Department of Veterans Affairs Medical Center, Salt Lake City, UT 84148, USA

**Keywords:** metastatic bladder cancer, urothelial carcinoma, geriatric patients, older adults, younger adults, chemotherapy, immunotherapy

## Abstract

**Simple Summary:**

Older adults with metastatic cancer of the urinary tract often do not receive optimal cancer treatments. Through our real-world study at a tertiary cancer center, we investigated the clinical characteristics, treatment patterns, and outcomes among older patients compared to younger adults receiving first-line systemic treatment. We found that older patients tend to be more suited to receiving immunotherapy and lower amounts of chemotherapy. When treated with regimens tailored to their overall health, they tolerate treatment as well as younger patients and experience similar life-prolonging benefits from these. Our results further reinforce that age alone is not a predictive factor for survival in patients who receive systemic treatment for advanced cancer. These findings suggest the need for appropriate treatment selection and tailored regimens for older adults with metastatic cancer of the urinary tract.

**Abstract:**

Metastatic urinary tract cancer (mUTC) is challenging to treat in older adults due to comorbidities. We compared the clinical courses of younger and older (≥70 years) adults with mUTC receiving first-line (1L) systemic therapy in a tertiary cancer center. Baseline clinical characteristics, treatments received, tolerability, and survival outcomes were analyzed. Among 212 patients (103 older vs. 109 younger), the older patients had lower hemoglobin at baseline (84% vs. 71%, *p* = 0.03), the majority were cisplatin-ineligible (74% vs. 45%, *p* < 0.001), received more immunotherapy-based treatments in the 1L (52% vs. 36%, *p* = 0.01), received fewer subsequent lines of treatment (median 0 vs. 1, *p* = 0.003), and had lower clinical trial participation (30% vs. 18%, *p* = 0.05) compared to the younger patients. When treated with 1L chemotherapy, older patients required more dose adjustments (53.4% vs. 23%, *p* = 0.001) and received fewer cycles of chemotherapy (median 4 vs. 5, *p*= 0.01). Older patients had similar OS (11.2 months vs. 14 months, *p* = 0.06) and similar rates of treatment-related severe toxicity and healthcare visits, independent of the type of systemic treatment received, compared to younger patients. We conclude that select older adults with mUTC can be safely treated with immunotherapy and risk-adjusted regimens of chemotherapy with tangible survival benefits.

## 1. Introduction

The treatment paradigm for metastatic urothelial carcinoma, which arises most commonly from the bladder, has evolved dramatically over the past decade. Chemotherapy has been the cornerstone for treating most patients with the disease [1]. Before the availability of antibody–drug conjugates, medically fit patients with good performance status, adequate renal function, and without significant comorbidities were treated with cisplatin-based chemotherapeutic regimens [1,2]. For cisplatin-ineligible patients, carboplatin-based regimens have been a viable therapeutic option [1]. Based on appreciable tolerability and demonstrable efficacy across multiple clinical trials, immune checkpoint inhibitors targeting programmed cell death protein 1 (PD-1) or its ligand (PD-L1) have emerged as a promising treatment option for patients who are not candidates to receive or tolerate chemotherapy [1,3,4]. Despite the encouraging efficacy and survival outcomes observed in multiple clinical trials, most treatment options do not offer durable remission and prolonged survival. More recently, the combination of enfortumab vedotin with pembrolizumab has shown greater efficacy and a more manageable safety profile, along with prolonged duration of response and overall survival [5].

The median age of diagnosis of bladder cancer is 73 years, with a notable correlation with a smoking history [6]. Therefore, most real-world patients with metastatic disease are older with substantial comorbidities and are not representative of the clinical trial patient population. Multiple studies have reported that older patients tend to have worse survival outcomes compared to younger patients with bladder cancer [7,8,9]. Age has been reported as a risk factor for chemotherapy-related toxicity across multiple tumor types [10]. Additionally, older patients with cancer have competing comorbidities and are often deemed unfit for chemotherapy [11]. Treatment decision-making is often challenging and complex for oncologists and is primarily driven by pre-treatment functional status and comorbidities [12,13].

Historically, older adults with cancer have been under-represented in clinical trials across multiple cancer types [14,15,16,17,18]. However, recent studies have reported an increase in the enrollment of older adults with cancer in clinical trials [19,20]. Despite the increased participation of these patients in clinical trials, there is a lack of randomized data for appropriate age-directed therapies and survival outcomes in older adults with bladder cancer. Therefore, an unmet need exists to characterize this patient population and treatment outcomes. Here, we describe the patient and disease characteristics, treatment patterns, treatment-related toxicities, and survival outcomes among older compared to younger adults receiving first-line (1L) systemic therapy for metastatic urinary tract cancer (mUTC) at a tertiary cancer care center. We included patients who received first-line systemic therapy for mUTC with the primary site of disease anywhere in the upper tract (renal pelvicalyceal system and ureter, excluding renal cell carcinoma), ureter, bladder, or urethra. We included urothelial and non-urothelial histology to have a comprehensive study focused on comparing patterns of clinical features and treatment outcomes in younger and older patients; the representation of the less common UTC histology seen in the real world thus complemented the representation of the less common primary sites [21].

## 2. Methods

### 2.1. Patient Selection

This IRB-approved retrospective study included patients receiving care between 2014 and 2023 at the Huntsman Cancer Institute, University of Utah, a tertiary care National Cancer Institute-designated Comprehensive Cancer Center. Eligible patients had a confirmed diagnosis of mUTC and received treatment with first-line systemic therapy with chemotherapy- or immunotherapy-based regimens. Patients who did not receive systemic therapy for mUTC or did not have treatment history available for review were excluded. Patients with localized or locally advanced disease were excluded. We abstracted disease and demographic data, including age at diagnosis, gender, race, smoking status, tumor location, tumor stage, histology, presence of de novo disease at diagnosis, prior definitive radiation or surgical treatment, receipt of prior chemotherapy in the localized or locally advanced setting, type of first-line systemic therapy received, and subsequent lines of treatment received, from the electronic medical records. In our dataset, the type of definitive surgery received by patients included radical cystectomy, partial cystectomy, urethrectomy, and ureterectomy. We categorized histology as: (1) urothelial for those with primarily urothelial histology, (2) neuroendocrine if there was any degree of neuroendocrine component due to treatment guidelines recommending systemic therapy regimens irrespective of degree of neuroendocrine histology, (3) adenocarcinoma for those with primary adenocarcinoma, and (4) squamous cell carcinoma for those with primary squamous cell carcinoma. We also collected comorbidity data (the Eastern Cooperative Oncology Group Performance Status [ECOG], hearing impairment, peripheral neuropathy, and heart failure) and laboratory variables (estimated glomerular filtration rate [eGFR], hemoglobin, absolute neutrophil count [ANC], absolute lymphocyte count [ALC], and platelet cell counts) before receiving first-line systemic therapy. Treatment-related data, such as the number of cycles of first-line therapy received, treatment-related dose reductions, cisplatin ineligibility, grade 3 or higher toxicity, and healthcare admissions, were also abstracted. Healthcare admissions included unexpected hospital admissions, emergency department (ED), and acute care visits. Routine visits, including radiology, infusion, and office visits, were excluded.

Patients were classified into two groups based on age at the receipt of first-line systemic therapy for mUTC. Patients aged 70 or older were categorized as the ‘older’ group, while those less than 70 years of age were included in the ‘younger’ group. We also performed a focused evaluation of patients who were 85 and older at the time of receipt of first-line systemic therapy for mUTC. For reporting survival outcomes and treatment-related outcomes, such as the incidence of treatment-related dose adjustments, treatment-related toxicity, the incidence of healthcare visits, and discharge disposition at the end of healthcare visits, patients receiving standard-of-care first-line chemotherapy-based or immunotherapy-based treatments were included, while those enrolled in clinical trials were excluded.

### 2.2. Outcomes

We compared the OS between the older and the younger patients. The OS was defined as the time from initiation of the first-line systemic therapy to death from any cause. Patients alive during the follow-up were censored to the date of the last clinic visit as documented in the patient’s medical record. In addition, we compared the progression-free survival (PFS) between the older and younger cohorts. The PFS was defined as from the start of first-line systemic therapy to investigator-assessed radiographic or clinical progression or death. Patients alive during the follow-up were censored at the last clinic visit, as documented in the patient’s medical record.

### 2.3. Statistical Analysis

Baseline characteristics were summarized using median and interquartile range (IQR) for continuous variables and count and percentage for categorical variables. We compared the baseline characteristics using the Student’s *t*-test performed for continuous variables and the chi-squared test for categorical variables. Statistical significance was determined at 95% confidence interval (CI). OS and PFS were summarized using the Kaplan–Meier estimate. The multivariate Cox proportional hazards model adjusted for the variables age, sex, primary site, smoking status, presence of visceral metastases, receipt of prior definitive therapy, cisplatin eligibility ECOG, and albumin were performed to assess the factors influencing OS. All statistical analyses were performed using R (v 4.3.3).

## 3. Results

Figure 1 describes the patient selection for the cohorts in our study.

### 3.1. Baseline Characteristics

A total of 212 patients met the eligibility criteria and were included in the study. The older cohort included 103 adults with mUTC, while the younger group had 109 patients. The median age of the overall cohort was 69 years (range: 31–92 years). The baseline demographic and clinical characteristics of the patients are shown in Table 1. Patients from both cohorts were predominantly male (84% older vs. 75% younger, *p* = 0.09), White (95% older vs. 93% younger, *p* = 0.11), had lower tract tumors (73% older vs. 74% younger, *p* = 0.93), had metachronous metastatic disease (69% older vs. 62% younger, *p* = 0.39), had similar rates of receipt of prior definitive surgery (50% older vs. 54% younger, *p* = 0.69), and had stage II or higher disease at initial diagnosis (83% older vs. 76% younger, *p* = 0.24). Older patients were predominantly non-smokers compared to younger patients (53% vs. 43%, *p* < 0.001). Patients had predominantly urothelial histology (94% older vs. 90% younger, *p* = 0.99). There was no significant difference between the two groups among patients who received prior chemotherapy with neoadjuvant or adjuvant intent (32% in the older vs. 37% in the younger, *p* = 0.57). Rates of receipt of definitive treatment of primary tumor (excluding non-muscle invasive bladder cancer) was similar between the two groups (60% in the older vs. 57% in the younger, *p* = 0.73). Rates of use of surgery or radiation-based definitive treatment of the primary tumor were similar in the two cohorts. In total, 14 patients received both definitive surgery and radiotherapy-based treatment of the primary tumor.

Older patients were more likely to receive immunotherapy-based treatment (52%), while younger patients received predominantly chemotherapy-based regimens as the first-line systemic therapy (64%) (*p* = 0.01). Compared to the younger patients, older patients had significantly higher hearing impairment (as documented in the electronic medical records at baseline) (36% vs. 17%, *p* < 0.002), were more likely to be anemic prior to starting first-line therapy (84% vs. 71%, *p* = 0.03), and showed a trend towards an eGFR of less than 60 mL/min prior to starting first-line systemic therapy compared to younger patients (58% older vs. 44% younger, *p* = 0.07). The number of subsequent lines of therapy received was significantly lower in the older group compared to the younger group (median (range): 0 (0–3) vs. 1 (0–5), *p* = 0.003). Older patients were more likely to be cisplatin-ineligible than younger patients (74% vs. 45%, *p* < 0.001). The clinical trial participation rate was significantly lower in older vs. younger patients. (30% vs. 18% respectively, *p* = 0.05).

### 3.2. Survival Outcomes

In total, 160 patients (72 older vs. 88 younger) who did not enroll in clinical trials were eligible for survival and toxicity assessments. The median time to follow-up for OS was 43.8 months (95% CI 24.5–61.2). The median OS for older adults with mUTC receiving first-line systemic therapy was shorter than that of the younger cohort but did not reach statistical significance (11.2 months vs. 14.0 months, *p* = 0.066, Figure 2A). On comparing patients with urothelial histology, there was no significant difference in OS between the older and younger cohort (12.9 months vs. 16.3 months, *p* = 0.15, Figure 2B). A multivariate analysis assessing various patient and disease-related factors associated with diminished OS in the entire cohort is shown in Table 2. An ECOG of 2 at the initiation of first-line systemic therapy (HR: 2.47, 95% CI 1.46–4.19, *p* = 0.0080), presence of visceral metastases at diagnosis (HR: 2.12, 95% CI 1.40–3.22, *p* = 0.0004), and the presence of upper tract disease (HR: 1.65, 95% CI 1.04–2.64, *p* = 0.03) were significantly associated with worse OS in the study population. Cisplatin eligibility was associated with improved OS (HR: 0.63, 95% CI 0.41–0.97, *p* = 0.03).

No significant difference in PFS was observed between older vs. younger patients with mUTC receiving first-line systemic therapy (4.77 vs. 6.67 months, *p* = 0.50, Figure 3A). The PFS of those with urothelial histology was not significantly different between older vs. younger patients (4.90 months vs. 6.87 months, *p* = 0.81, Figure 3B). When interrogated by treatment arms, PFS continued to be similar between the younger vs. older patients: chemotherapy (7.17 vs. 4.20 months, *p* = 0.33) and immunotherapy (4.8 vs. 4.9 months, *p* = 0.94), (Appendix A, respectively). The OS and PFS stratified by histological subtypes were insignificant; *p* = 0.37 and *p* = 0.12, respectively. (Appendix A).

### 3.3. Treatment-Related Outcomes

In the overall cohort, 108 patients received a chemotherapy-based regimen, and 52 patients received an immunotherapy-based regimen as the first-line therapy. Immunotherapy-based treatment received by patients included atezolizumab (n = 6), nivolumab (n = 5), and pembrolizumab (n = 41). Among patients receiving chemotherapy, 38/108 (35.1%) patients received dose adjustments at any time in treatment, 6/38 (15.7%) had pre-emptive dose reduction, 29/38 (76.3%) had dose adjustments after cycle 1, and 3/38 (7.9%) patients received both pre-emptive and subsequent dose reductions. Among the 108 patients receiving first-line chemotherapy, 43 were older patients, while 65 were younger patients. Compared to younger patients, the older group had significantly more patients who received overall dose adjustments: 23/43 (53.4%) vs. 15/65 (23.0%), *p* = 0.001. Further, the median number of cycles of chemotherapy was significantly lower in the older patients compared to younger patients (median; range 4 (0–6) vs. 5 (1–12), *p* = 0.01) [Table 1].

There was no significant difference between the incidence of grade 3 or higher toxicities observed among older patients with mUTC compared to younger patients with mUTC independent of the type of first-line treatment received (44.1% older vs. 38.4% younger, *p* = 0.55 and 24.1% vs. 26.0%, *p* = 0.88 in the chemotherapy and immunotherapy-based treatment arms, respectively). Table 3 summarizes grade 3 or higher treatment-related toxicities observed in younger vs. older patients with mUTC by the type of first-line systemic therapy received. Among older patients receiving chemotherapy-based treatment, 44.1% experienced grade 3 or higher treatment-related toxicity. Among the grade 3 or higher toxicities, myelosuppression-induced complications (15/19) were most common, followed by hepatotoxicity (2/19), diarrhea (1/19), and unspecified intolerance (1/19). The frequency of younger mUTC patients experiencing a grade 3 or higher chemotherapy-related toxicity was 38.4%. The most commonly encountered grade 3 or higher toxicities were myelosuppression (20/25), fatigue (2/25), arterial thrombosis (1/25), tinnitus (1/25), and unspecified intolerance (1/25). Among those who received first-line immunotherapy-based treatment, 25% of patients experienced grade 3 or higher treatment-related toxicity. Of these 13/52 patients, seven were older, while six were younger. Among older patients, the most commonly experienced grade 3 or higher treatment-related adverse events were colitis (2/7), skin rash (2/7), elevated liver enzymes (2/7), and nephritis (1/7). Myositis (2/6), elevated liver enzymes (2/7), skin rash (1/6), and colitis (1/6) were most common among younger patients.

### 3.4. Treatment-Associated Healthcare Visits

In the overall cohort, the number of patients requiring at least one healthcare visit for treatment-related complications was 35/72 (48.6%) in older vs. 46/88 (52.2%) among younger patients, independent of the type of first-line treatment received. When subdivided by treatment groups, 21/43 (48.8%) of older and 35/65 (53.8%) younger patients in the chemotherapy cohort required at least one healthcare visit. In comparison, 14/29 (48.2%) of older and 11/23 (47/8%) younger patients in the immunotherapy cohort needed at least one healthcare visit. Among the patients requiring at least one healthcare visit for treatment-related complications, the total number of ED visits and hospital admissions cumulatively were 40 in older vs. 62 in younger and 36 in older vs. 65 in younger patients in the chemotherapy and immunotherapy cohorts, respectively. Based on the type of first-line systemic therapy received, no significant differences were observed in the median number of ED and hospital admissions between the two cohorts in the chemotherapy and the immunotherapy cohorts. Table 4 summarizes treatment-complication-related healthcare visits between younger vs. older patients by treatment group.

### 3.5. Discharge Disposition

Of the 81 adults requiring healthcare visits for treatment-related complications, 35 were older, while 46 were younger. Table 5 summarizes the differences in the discharge disposition between the older vs. younger patients with mUTC receiving first-line systemic therapy. The frequency of patients who could be discharged home with self-care was comparable between the two groups (82.8% [29/35] in older vs. 77.7% [35/45] in younger, *p* = 0.57). The rates of skilled nursing home disposition were numerically different between the two groups (14.2% [5/35] older vs. 8.8% [4/45] younger, *p* = 0.45). Overall, there was no significant difference in the pattern of discharge disposition among older and younger patients with mUTC.

### 3.6. Patients Aged 85 Years and Above

Our study included eight patients aged 85 years and above at the time of receipt of first-line systemic therapy for mUTC. The majority of these patients were male (7/8), and all were White. One of the eight patients had synchronous metastatic disease. Three of the eight patients received prior chemotherapy in the localized or locally advanced setting. Seven of the eight patients received a standard of care regimen outside of clinical trials. Four patients received chemotherapy-based regimens, while four received immunotherapy-based regimens as first-line systemic therapy. The median number of subsequent lines of therapy received in these seven patients was 0 (0–1). The median OS among these seven patients was 3.9 months (range 0.36–28.46 months).

## 4. Discussion

Our study reveals that patients 70 years and older, when compared to patients younger than 70 years, receiving first-line systemic therapy for mUTC, tend to be cis-ineligible and have reduced myeloid reserve. Clinical trial participation and the number of lines of treatment were notably lower in this cohort. When treated with preemptive and reactive dose-adjusted chemotherapy and immunotherapy regimens, older patients experienced appreciable benefits with comparable OS or PFS compared to younger patients with mUTC receiving first-line systemic therapy. These tailored regimens are noted to be well tolerated with similar degrees of severe side effects and comparable rates of use of acute healthcare facilities to manage treatment-related adverse effects.

Our study represents the real-world population of patients presenting for treatment of their metastatic urinary tract cancer at a tertiary cancer care center. We included non-bladder disease sites and non-urothelial histology to represent the real-world population. Although there were no significant differences between the older versus younger patients in the primary site of disease and proportion of urothelial histology, we note that adenocarcinoma and squamous cell carcinoma are seen predominantly in the younger group (Table 1).

Multiple studies have highlighted the influence of age in treatment selection among patients with metastatic bladder cancer [22,23]. For instance, Galsky et al. showed that less than 50% of patients aged 65 years and above received first-line chemotherapy [22]. Similarly, Sonpavde et al. showed that cisplatin was mostly prescribed among patients under 70 years, while those over 70 either received non-cisplatin-based treatments or no treatment [23]. Despite the evidence on comparable safety and efficacy of cisplatin-based therapies in medically fit cisplatin-eligible patients [24], little has changed in real-world treatment patterns in recent times. For instance, in a large population-based Flatiron study, Morgans et al. reported that among patients receiving immune checkpoint inhibitors in the first line or second line of therapy, more than 80% of patients were aged 65 years and above [25]. Immune checkpoint inhibitors have demonstrated excellent tolerability, with acceptable efficacy among patients with metastatic bladder cancer across multiple clinical trials [3,4,26]. Hence, there exists an inherent bias toward prescribing non-chemotherapeutic regimens for older patients among treating clinicians [23].

Treatment decision-making for older patients with mUTC is complex and driven by multiple factors, such as patient preference, physician’s assessment for treatment tolerability, and patient frailty [27]. Though age may be associated with increased frailty and treatment toxicity, it does not constitute a reliable prognostic factor [28]. Our study showed that older patients had comparable outcomes to their younger counterparts regardless of the systemic treatment received. Our study further reiterated that selected older patients can derive benefits from first-line chemotherapy with improved tolerability given appropriate pre-treatment assessment, preemptive dose reductions, and management of treatment toxicities through appropriate treatment dose adjustments. Though limited evidence exists in the context of bladder malignancy, geriatric assessment tools could be used to quantify frailty further [29,30]. Geriatric assessment tools such as the Cancer and Aging Research Group Chemotherapy Toxicity Tool (CARG-TT) [10] and the Chemotherapy Risk Assessment for High-Age Patients (CRASH) [31] can be used to assess the risk of chemotherapy-associated toxicity in older patients with mUTC. Further studies are needed to validate the clinical utility of these assessment tools in treatment selection regarding novel therapies such as antibody–drug conjugates. The toxicities observed with some antibody–drug conjugates and immunotherapy differ significantly from cytotoxic chemotherapy, demanding the need for a toxicity prediction tool tailored to novel regimens. Additionally, there is a substantial need for the adoption of oncology practice guidelines regarding the incorporation of age-based assessment tools in clinical practice to make informed treatment decisions for the older patient population with mUTC. In a randomized controlled clinical trial into Geriatric Assessment-Driven Intervention (GAIN) on Chemotherapy-Related Toxic Effects—with 605 older adults looking at the incidence of reduced Grade 3 or higher chemotherapy-related toxicity with GAIN intervention—a 10.1% reduction of Grade 3 or higher toxicity was noted with geriatric assessment [32].

In a cluster randomized study that evaluated the effect of geriatric assessment and management on the toxic effects of cancer treatment (GAP70+), a lower proportion of patients in the intervention group had grade 3–5 toxic effects. Patients in the intervention group had fewer falls over three months and had more medications discontinued. Age-sensitive care for older patients with advanced cancer reduced the burden of serious toxicity from cancer treatment while enabling the clinical benefit of systemic therapy seen in younger patients [33].

Our study has limitations, including its retrospective nature, small sample size, single institutional experience, and non-randomization of patients to receive first-line therapy. Moreover, we only included patients with mUTC who received first-line systemic therapy for their disease, which creates a selection bias and a possible inherent physician bias in selecting medically fit patients to receive chemotherapy. Further, we would like to indicate that PD-L1 assessment was not used in determining eligibility for treatment with immunotherapy-based regimens among mUTC patients. Hence, our study lacks a correlation between PD-L1 expression and the clinical outcomes among younger and older patients receiving immunotherapy-based regimens. We also note the predominantly White patient population, which limits the representation within our study results of other races. Our study predates the use of antibody–drug conjugate plus immunotherapy combination in the first-line setting for metastatic urothelial carcinoma. Notwithstanding these limitations, our study provides real-world evidence on the age-based survival estimates, treatment patterns, and treatment-related toxicities among older patients with mUTC receiving care at a tertiary-level academic cancer center.

## 5. Conclusions

While reduced organ function reserve is more prevalent in older adults with mUTC, risk-adjusted therapeutic regimens can be used safely, and survival benefit can be derived from treatment. Our results further reinforce that age alone is not a predictive factor for survival in patients who receive systemic treatment for cancer. Our findings suggest the need for appropriate patient selection and tailored regimens and provide information for patient counseling and clinical trial design for the aging population with metastatic cancer of the urinary tract.

## Figures and Tables

**Figure 1 cancers-16-02143-f001:**
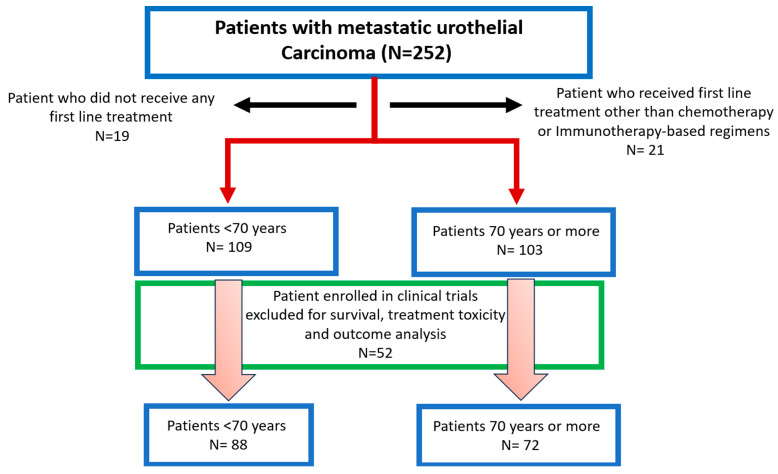
CONSORT diagram.

**Figure 2 cancers-16-02143-f002:**
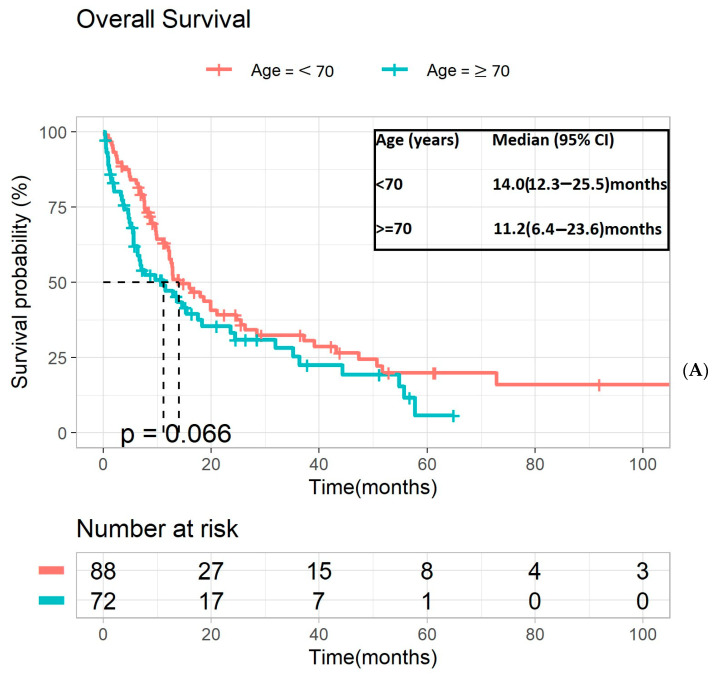
Kaplan–Meier analysis of OS of younger versus older patients receiving standard-of-care 1L therapy (**A**) for all histology subtypes, (**B**) for urothelial histology. OS—overall survival; 1L—first line.

**Figure 3 cancers-16-02143-f003:**
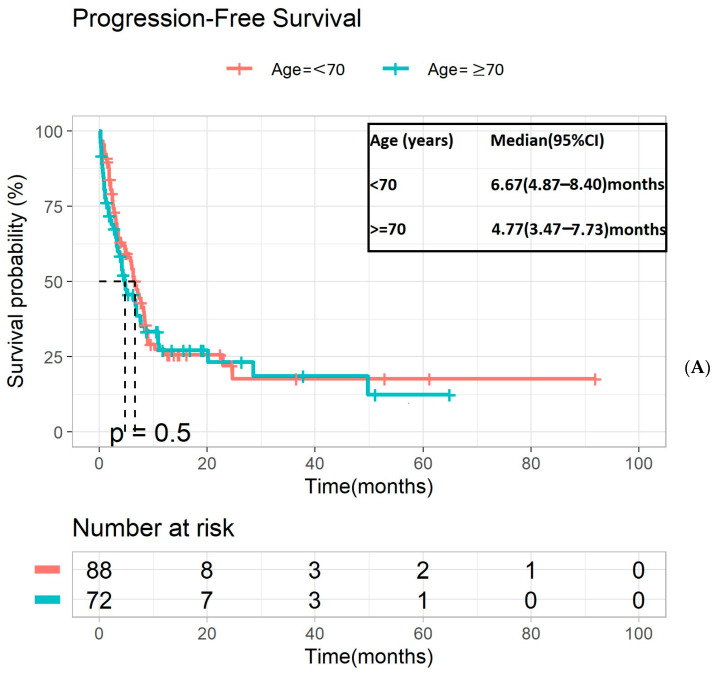
Kaplan–Meier analysis for PFS of younger versus older patients receiving standard of care 1L therapy (**A**) for all histology subtypes, (**B**) for urothelial histology. PFS—progression-free survival; 1L—first line.

**Table 1 cancers-16-02143-t001:** Baseline demographics of older vs. younger patients with metastatic urinary tract cancer receiving first-line systemic therapy (1L Rx). Ca—carcinoma; RT—radiation therapy; CRT—chemoradiation therapy.

Variables	<70 Years (*n* = 109)	≥70 Years (*n* = 103)	*p* Value
Clinical Characteristics			
1L Treatment Received n (%)			0.01
Chemotherapy	70/109 (64%)	49/103 (48%)	
Immunotherapy	39/109 (36%)	54/103 (52%)	
Gender; n (%)			0.09
Male	82/109 (75%)	87/103 (84%)	
Female	27/109 (25%)	16/103 (16%)	
Race; n (%)			0.11
White	101/109 (93%)	98/103 (95%)	
Non-white	8/109 (7%)	5/103 (5%)	
Smoking Status; n (%)			<0.0001
Yes	61/107 (57%)	48/102 (47%)	
No	46/107 (43%)	54/102 (53%)	
Tumor Location; n (%)			0.93
Lower tract	81/109 (74%)	75/103 (73%)	
Upper tract	28/109 (26%)	28/103 (27%)	
De novo metastatic disease at diagnosis; n (%)			0.39
Yes	41/109 (38%)	32/103 (31%)	
No	68/109 (62%)	71/103 (69%)	
Tumor Stage; n (%)			0.24
Stage less than II	23/98 (24%)	16/96 (17%)	
Stage II or more	75/98 (76%)	80/96 (83%)	
Tumor Histology; n (%)			0.99
Urothelial carcinoma	98/109 (90%)	97/103 (94%)	
Squamous cell carcinoma	2/109 (2%)	1/103 (1%)	
Neuroendocrine carcinoma	5/109 (5%)	5/103 (5%)	
Adenocarcinoma	4/109 (3%)	0	
Prior definitive treatment of primary with surgery, RT, CRT *	62/109 (57%)	62/103 (60%)	0.73
Prior definitive surgery **	59/109 (54%)	52/103 (50%)	0.69
Prior definitive RT or CRT (n = 27) ***	12/109 (11%)	12/103 (15%)	0.57
Prior Chemotherapy (neoadjuvant/adjuvant intent)			0.57
Yes	40/109 (37%)	33/103 (32%)	
No	69/109 (63%)	70/103 (68%)	
*Lab Characteristics*			
ECOG at 1L Rx start			0.75
ECOG (0–1)	75/89 (84%)	79/97 (81%)	
ECOG (2–4)	14/89 (16%)	18/97 (19%)	
eGFR at 1L Rx start			0.07
<60 mL/min	40/90 (44%)	56/97 (58%)	
≥60 mL/min	50/90 (56%)	41/97 (42%)	
Albumin at 1L Rx start			0.38
<3.5 mg/dL	17/95 (18%)	23/100 (23%)	
≥3.5 mg/dL	78/95 (82%)	77/100 (77%)	
Hemoglobin at 1L Rx start ****			0.03
Below normal	70/98 (71%)	85/101 (84%)	
Normal or above	28/98 (29%)	16/101 (12%)	
Absolute lymphocyte count; median (range)	1.25 (0.32–6.1)	1.2 (0.27–9.1)	0.79
Absolute neutrophil count; median (range)	4.86 (2.19–17.07)	5.19 (1.96–16.63)	0.88
Platelet at 1L Rx start			0.16
<159 × 10^9^/L	6//98 (6%)	12/101 (12%)	
439 × 10^9^/L	92/98 (94%)	89/101 (88%)	
Clinical Trial Participation			0.05
Yes	20/109 (18%)	31/103 (30%)	
No	89/109 (82%)	72/103 (70%)	
Any hearing impairment			<0.002
Yes	19/109 (17%)	37/103 (36%)	
No	90/109 (83%)	66/103 (64%)	
Any grade neuropathy			0.4
Yes	17/109 (16%)	12/103 (12%)	
No	92/109 (84%)	91/103 (88%)	
Heart failure			0.7
Yes	7/109 (6%)	8/103 (8%)	
No	102/109 (94%)	95/103 (92%)	
Cisplatin eligibility			<0.001
Yes	48/88 (55%)	19/72 (26%)	
No	40/88 (45%)	53/72 (74%)	
Subsequent lines of therapy received; median (range)	1 (0–5)	0 (0–3)	0.003
Number of cycles of chemotherapy (n = 108) *****;			0.01
median (range)	5 (1–12)	4 (0–6)	
Number of cycles of immunotherapy received; median (range)	3 (1–35)	5 (1–22)	0.55
Death events	58/109 (53%)	50/103 (49%)	0.59

* Excluding treatments for non-muscle invasive bladder cancer. ** Partial cystectomy (PC), radical cystectomy (RC), nephroureterectomy (NU), urethrectomy (Uth), ureterectomy (Ur). Age < 70: 6 PC, 36 RC, 14 NU, 2 Uth, 1 Ur, Age ≥ 70 years: 2 PC, 36 RC, 12 RNU, 1 Uth, 1 Ur. *** Age < 70 2RT, 10 CRT, Age ≥ 70 2RT, 13 CRT. **** Normal range for hemoglobin in male: 14.8–17.8 g/dL, female: 12.6–15.9 g/dL. ***** includes non-clinical trial patients: <70 years- 65 and >=70 years: 43.

**Table 2 cancers-16-02143-t002:** Multivariate analysis using Cox proportional hazards for OS of patients with metastatic urinary tract cancer receiving 1L systemic therapy. OS—overall survival; 1L—first line; ECOG—Eastern Cooperative Oncology Group Performance status; mUTC—metastatic urinary tract cancer.

Baseline Variables at Start of 1L Systemic Therapy	OS Hazard Ratio (95% Confidence Interval, *p* Value)
Age at diagnosis of mUTC (≥70)	1.28 (0.84–1.94, 0.25)
Sex (male)	0.80 (0.48–1.32, 0.38)
Tumor location (upper tract)	1.65 (1.04–2.64, 0.03)
Smoking status (yes)	0.80 (0.54–1.20, 0.28)
Presence of visceral metastases	2.12 (1.40–3.22, 0.0004)
Presence of De novo disease at diagnosis (yes)	1.26 (0.73–2.18, 0.41)
Receipt of prior definitive surgery or radiation	0.59 (0.34–1.10, 0.05)
Cisplatin eligibility (yes)	0.63 (0.41–0.97, 0.03)
ECOG of 2 or higher	2.47 (1.46–4.19, 0.0008)
Albumin (normal ≥ 3.5 mg/dL)	0.80 (0.52–1.22, 0.29)

**Table 3 cancers-16-02143-t003:** Grade 3 or higher treatment-related toxicities by the type of first-line systemic therapy received.

	<70 Years	≥70 Years	*p* Value
Chemotherapy (n = 108); n (%)	25/65 (38.4)	19/43 (44.1)	0.55
Immunotherapy (n = 52); n (%)	6/23 (26.0)	7/29 (24.1)	0.88

**Table 4 cancers-16-02143-t004:** Treatment related healthcare visits. (ED—emergency department).

	<70 Years	≥70 Years	*p* Value
**Chemotherapy (n = 108); median (range)**			
ED visits	0 (0–8)	0 (0–3)	0.94
Hospitalization	0 (0–4)	0 (0–3)	0.36
**Immunotherapy (n = 52); median (range)**			
ED visits	0 (0–7)	0 (0–2)	0.54
Hospitalization	0 (0–4)	0 (0–2)	0.90

**Table 5 cancers-16-02143-t005:** Discharge disposition.

	<70 Years (n = 45)	≥70 Years (n = 35)	*p* Value
Home with self-care; n (%)	35/45 (77.7)	29/35 (82.8)	0.57
Home with home health care; n (%)	18/45 (40.0)	8/35 (22.8)	0.10
Home Hospice; n (%)	5/45 (11.1)	7/35 (20.0)	0.27
Skilled Nursing home; n (%)	4/45 (8.8)	5/35 (14.2)	0.45
Death; n (%)	3/45 (6.6)	1/35 (2.8)	0.43

## Data Availability

The datasets presented in this article are not readily available, as some of the data include patients involved in clinical trials. Requests to access the datasets should be directed to the corresponding author, Sumati Gupta.

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
