# Peer review of "Treatment Patterns and Outcomes by Age in Metastatic Urinary Tract Cancer: A Retrospective Tertiary Cancer Center Analysis"

_cancers, 2024, doi:10.3390/cancers16112143_

Round 1

Reviewer 1 Report

Comments and Suggestions for Authors

The authors should be congratulated for their work. Specifically, the study aimed to describe the patient and disease characteristics, treatment patterns, treatment-related toxicities, and survival outcomes among older (more of 70 years) compared to younger adults receiving first-line (1L) systemic therapy for metastatic urinary tract cancer (mUTC) at a tertiary cancer care center. The study relied on 202 mUTC patients enrolled in a single institution within 10 years. The population represents one of the most contemporary observed (2014-2023).

The results are intriguing. Specifically, the most intriguing part and not expected is that older patients required more dose adjustments and received fewer cycles of chemotherapy. These observations don't result in different OS (11.2 months vs. 14 months, p=0.06), or related side effects. This is a very important quality of care indicator. 

However, several points should be addressed:

- Are there patients who receive TMT? Which proportion? This treatment is established over time, both in urothelial and non-urothelial histology as well as organ confined and non-confined as well as older and younger (PMID= 38494989).

- Despite the endpoints being OS and PFS, it should be acknowledged that those patients could die from cancer-specific mortality, specifically in a metastatic setting (PMID= 38199879). How many events did the authors have? Bladder cancer is highly more lethal than other urological cancers, and so is the UTUC (PMID= 34330651).  They should be addressed differently. Specifically, the role of CSM cannot be ignored and should be reported more than OS.

- It is not very clear and informative why the patients with non-urothelial histology were not excluded from the analyses (only 17 patients in total). Moreover, there is no rational to report it in the multivariable model.

- The multivariable model, which appears overfitted, required a further adjustment for clinically meaningful variables such as: first, sex (EAU guidelines reported a potentially different effect of chemo according to sex ) that should be accounted for; second, tumor location, third, smocking status.  Specifically, It would be strongly advised to add a subgroup analysis to look at whether the difference recorded in the current manuscript applied in UTUC as well as UCUB cancer. Moreover, the eGFR and the cisplatin receipt said the same thing. It should be rational to administer the cisplatin only in patients whose eGFR is enough performant. 

- Any ancillary information on microbiota (PMID 37289890)?

Author Response

The authors should be congratulated for their work. Specifically, the study aimed to describe the patient and disease characteristics, treatment patterns, treatment-related toxicities, and survival outcomes among older (more of 70 years) compared to younger adults receiving first-line (1L) systemic therapy for metastatic urinary tract cancer (mUTC) at a tertiary cancer care center. The study relied on 202 mUTC patients enrolled in a single institution within 10 years. The population represents one of the most contemporary observed (2014-2023).

The results are intriguing. Specifically, the most intriguing part and not expected is that older patients required more dose adjustments and received fewer cycles of chemotherapy. These observations don't result in different OS (11.2 months vs. 14 months, p=0.06), or related side effects. This is a very important quality of care indicator. 

However, several points should be addressed:

1)  Are there patients who receive TMT? Which proportion? This treatment is established over time, both in urothelial and non-urothelial histology as well as organ confined and non-confined as well as older and younger (PMID= 38494989).

We would like to thank the reviewers for their comment. Yes, there were patients who received trimodailty therapy in our dataset. Rates of receipt of definitive treatment with  surgery, radiation and chemoradiation therapy was similar between the two groups (60% older vs 57% younger, p=0.73). [Please refer to line 160-165 and table 1]

2)  Despite the endpoints being OS and PFS, it should be acknowledged that those patients could die from cancer-specific mortality, specifically in a metastatic setting (PMID= 38199879). How many events did the authors have? Bladder cancer is highly more lethal than other urological cancers, and so is the UTUC (PMID= 34330651).  They should be addressed differently. Specifically, the role of CSM cannot be ignored and should be reported more than OS.

We thank the reviewer for pointing this out. Please note that this study involves patients with metastatic urinary tract cancer and hence survival is mostly limited by disease. Calculating cancer specific mortality would be more apt in patients with earlier stages of disease Exploring the cancer-specific mortality in context of UTUC and UC may be explored in earlier stages of disease.

3) It is not very clear and informative why the patients with non-urothelial histology were not excluded from the analyses (only 17 patients in total). Moreover, there is no rational to report it in the multivariable model.

Through our real-world study, we performed a comprehensive analysis comparing the difference in outcomes between young and old patients with bladder cancer irrespective of histology and inclusion of non-urothelial histological subtypes is reflective of the real-world patterns of disease management burden by age.  However, given the difference in clinical outcomes for patients with non-urothelial histology, we assessed OS by histology to account for possible bias.( supplemental figure)

4) The multivariable model, which appears overfitted, required a further adjustment for clinically meaningful variables such as: first, sex (EAU guidelines reported a potentially different effect of chemo according to sex ) that should be accounted for; second, tumor location, third, smocking status.  Specifically, It would be strongly advised to add a subgroup analysis to look at whether the difference recorded in the current manuscript applied in UTUC as well as UCUB cancer. Moreover, the eGFR and the cisplatin receipt said the same thing. It should be rational to administer the cisplatin only in patients whose eGFR is enough performant. 

We would like to thank the reviewer for their suggestion. We changed our multivariable model to include age, sex, tumor location, presence of visceral metastases, presence of de novo disease at diagnosis, and receipt of prior definitive therapy of primary tumor. We found that the presence of upper tract disease was significant for worse OS among older patients (HR: 1.65, 95% CI 1.04-2.64, p=0.03). We agree with the reviewer that eGFR and cisplatin eligibility are variables representative of the ability of patients to receive chemotherapy and hence excluded eGFR from the multivariable model.  (Please refer to table 2)

5) Any ancillary information on microbiota (PMID 37289890)?

Ancillary information on microbiota would be beyond the scope of our study as our data set does not include any data pertaining to the microbiota

Reviewer 2 Report

Comments and Suggestions for Authors

This is a retrospective study on 212 patients with mUTC carcinoma, their treatment choices, and derived results in term of OS and PFS. Comorbidities and therapy related complications are also reviewed. The study includes patients from 2014-2023. Patients were divided by age > or < 70years. Younger patients were mostly treated by systemic chemotherapy. Older patients were mostly treated by systemic immunotherapy.

Comments and suggestions for improvement/clarification:

1.        The study is retrospective but accumulates a good number of cases (n=2012) to allow adequate conclusions and has some interesting results since the manuscript may provide a quick approach to mUTC by the oncologist using age in the primary assessment and to share expectations with the patient.

2.        The study includes neuroendocrine tumors and tumors with focal neuroendocrine differentiation. This may be misguiding, and the related results would probably need to be presented separately.  This is because this group of patients are treated by systemic chemo with good results if the patient is fit for the treatment.  The same may be of interest for the other histotypes. Then, I would suggest adding a table including the analysis and main results split by the 4 tumor histotypes included in the series. Tnhis comparison my be of interest to the reader

3.        Concerning immunotherapy there were all treated with the same protocol?  What was the role of PD-L1 assessment in this treatment? I would suggest a some more information on that, in text and in the discussion section…

4.        I guess the reader would expect a bit more information on those patients participating in clinical trials. What was the line of therapy, the single vs. combination agents, results in term of OS and PFS, toxicities and so on…. Perhaps a table on this would it be of interest to add.

Author Response

This is a retrospective study on 212 patients with mUTC carcinoma, their treatment choices, and derived results in term of OS and PFS. Comorbidities and therapy related complications are also reviewed. The study includes patients from 2014-2023. Patients were divided by age > or < 70years. Younger patients were mostly treated by systemic chemotherapy. Older patients were mostly treated by systemic immunotherapy.

Comments and suggestions for improvement/clarification:

1)  The study is retrospective but accumulates a good number of cases (n=2012) to allow adequate conclusions and has some interesting results since the manuscript may provide a quick approach to mUTC by the oncologist using age in the primary assessment and to share expectations with the patient.

2 ) The study includes neuroendocrine tumors and tumors with focal neuroendocrine differentiation. This may be misguiding, and the related results would probably need to be presented separately.  This is because this group of patients are treated by systemic chemo with good results if the patient is fit for the treatment.  The same may be of interest for the other histotypes. Then, I would suggest adding a table including the analysis and main results split by the 4 tumor histotypes included in the series. This comparison may be of interest to the reader

We thank the reviewer for their comments. Our study focuses on comparing clinical comparing between older and younger patients receiving treatment for mUTC. 17 patients had non-urothelial histology. Given very few patients with non-urothelial histology, inclusion of analysis based on histological subtypes would be beyond the scope of our study. This represents a real-world study and we included all histological subtypes to remain true to the real world pattern of disease management burden by age.

 3 ) Concerning immunotherapy there were all treated with the same protocol?  What was the role of PD-L1 assessment in this treatment? I would suggest a some more information on that, in text and in the discussion section…

We thank the reviewer for their comment. Most patients underwent tumor genomic testing  prior to the initiation of systemic therapy. Most patients underwent PD-L1 testing. However, PDL1 expression was not necessarily a criteria in selecting patients to undergo treatment with IO-based therapy.  We have mentioned the same in the discussion. (Please refer to lines 364-368)

4) I guess the reader would expect a bit more information on those patients participating in clinical trials. What was the line of therapy, the single vs. combination agents, results in term of OS and PFS, toxicities and so on…. Perhaps a table on this would it be of interest to add.

Patient in clinical trials were excluded from analysis pertaining to treatment-related outcomes as some of the trials have not been reported yet.

Reviewer 3 Report

Comments and Suggestions for Authors

It is a retrospective study comparing the efficacy and safety of 1L chemotherapy in patients over and below 70 years. The study desing is adequate although there is lack of important information.

1) Did the patients have prior surgical therapy? If yes, what type?

2) A stratification based on the tumor location would be beneficial (bladder?, ureter? renal pelvis?)

3) The significance of preoperative chemotherapy in combination and the postoperative application may be evaluated.

Author Response

It is a retrospective study comparing the efficacy and safety of 1L chemotherapy in patients over and below 70 years. The study design is adequate although there is lack of important information.

1) Did the patients have prior surgical therapy? If yes, what type?

Yes. 101 patients of 212 patients underwent prior definitive surgical therapy. Prior surgical therapies included radical cystectomy, partial cystectomy, urethrectomy,nephroureterectomy and ureterectomy. Among younger patients, 59/104 underwent the following definitive surgeries:  6 -partial cystectomy, 36 - radical cystectomy, 2- urethrectomy, 14 – nephroureterectomy, and 1-  ureterectomy. Among older patients, 52 of 103 underwent the following definitive surgeries: 2-partial cystectomy, 36 - radical cystectomy, 1- urethrectomy, 12 - nephroureterectomy, and 1-  ureterectomy. (Please refer to lines 93-95 in methods and see table 1)

2) A stratification based on the tumor location would be beneficial (bladder?, ureter? renal pelvis?)

We would like to thank the reviewer for their suggestion. As per the reviewer’s suggestion, we changed our multivariable model for OS to include tumor location and found that the presence of upper tract disease was significant for worse OS   (HR: 1.65, 95% CI 1.04-2.64, p=0.03). (Please refer to lines 199-201 and table 2)

3) The significance of preoperative chemotherapy in combination and the postoperative application may be evaluated.

We thank the reviewer for their comment. Receipt of prior chemotherapy in the adjuvant or neoadjuvant setting was not significantly different between older and young patients (32% vs 37%, p=0.57). (Please refer to table 1).    

Round 2

Reviewer 1 Report

Comments and Suggestions for Authors

The authors addressed properly my suggestions

Reviewer 2 Report

Comments and Suggestions for Authors

The authors have properly addressed the sugestions made by this reviewer. 

Comments on the Quality of English Language

English is fine. 

Reviewer 3 Report

Comments and Suggestions for Authors

All the required revisions were performed. I have no additional comments. It may be considered for publication.